# Prevalence and Risk Factors of Musculoskeletal Disorders in Basketball Players: Systematic Review and Meta-Analysis

**DOI:** 10.3390/healthcare11081190

**Published:** 2023-04-21

**Authors:** Silvia Cristina de Carvalho Borges, Carolina Rodrigues Mendonça, Regina Márcia Ferreira Silva, Alberto De Vitta, Matias Noll

**Affiliations:** 1Health Science, Universidade Federal de Goiás, Goiânia 74001-970, Brazil; 2Departament of Education, Instituto Federal Goiano, Ceres 76300-000, Brazil; 3Departament of Physical Therapy, Centro Universitário das Faculdades Integradas de Ourinhos, Ourinhos 19909-100, Brazil

**Keywords:** back pain, athletes, sport

## Abstract

Musculoskeletal disorders characteristically induce pain and limitations in mobility, ability, and overall functioning. In athletes, including basketball players, disorders such as back pain, postural changes, and spinal injuries are common. This systematic review aimed to evaluate the prevalence of back pain and musculoskeletal disorders in basketball players and ascertain the associated factors. Methods: The Embase, PubMed, and Scopus databases were searched for studies published in English without a time limit. Using STATA, meta-analyses were performed to estimate the prevalence of pain and musculoskeletal disorders of the back and spine. Results: Of the 4135 articles identified, 33 studies were included in this review, with 27 studies included in the meta-analysis. Of these, 21 were used for the meta-analysis of back pain, 6 articles were used for the meta-analysis of spinal injury, and 2 studies were used for the meta-analysis of postural changes. The overall prevalence of back pain was 43% [95% CI, −1% to 88%]; of these, the prevalence of neck pain was 36% [95% CI, 22–50%], the prevalence of back pain was 16% [95% CI, 4–28%], the prevalence of low back pain was 26% [95% CI, 16–37%], the prevalence of thoracic spine pain was 6% [95% CI, 3–9%]. The combined prevalence of spinal injury and spondylolysis was 10% [95% CI, 4–15%], with a prevalence of spondylolysis of 14% [95% CI, 0.1–27%]. The combined prevalence of hyperkyphosis and hyperlordosis was 30% [95% CI, 9–51%]. In conclusion, we found a high prevalence of neck pain, followed by low back pain and back pain, in basketball players. Thus, prevention programs are important to improve health and sports performance.

## 1. Introduction

Approximately 1.71 billion people worldwide suffer from musculoskeletal disorders that affect multiple areas or body systems, such as joints, bones, muscles, and spine; they are characterized by pain and limitations in mobility, ability, and overall level of functioning [1,2,3]; and necessitate rehabilitation [1]. Unlike that in the general population, musculoskeletal disorders can be exacerbated in athletes. Athletes frequently experience musculoskeletal disorders, with back pain being one of the most common symptoms [4,5,6]. Several sports are characterized by the need for specific movements, which may result in excessive spinal stress [7].

A study of elite German athletes from different sports reported that 77% of the athletes reported low back pain, followed by neck pain (63%) and thoracic spine pain (46%) [8]. Another study of 1114 elite German athletes from various sports reported an 89% prevalence of low back pain [9]. In another study of athletes from Finland, 46% of the basketball players who participated in the study had low back pain [10].

Basketball players frequently suffer from low back and neck pain [7,8,9,10,11], which confers a high risk for spinal injuries [7], and lumbar spine injuries are common in basketball players [12,13]. A study that longitudinally evaluated all injuries in the National Basketball Association (NBA) players over a 17-year period found that 10.2% of all injuries involved the lumbar spine and 0.9% of all injuries were due to lumbar disk degeneration [4].

Moreover, postural changes are common in players of various sports due to the repetitive and unilateral overload of the body during sports practice [14,15]. Furthermore, the trend of spinal curvature changes in basketball players, when compared to that in non-athletes, suggests an effect of regular basketball training on the degree of curvature of thoracic kyphosis and lumbar lordosis [16].

Basketball is an asymmetric sport that involves repetitive unilateral movements [15]. Therefore, the practice of this sport may promote pain and/or injury to the spine due to the number of throws and dribbles during a practice session or game [15]. Therefore, it is necessary to determine the prevalence of postural changes and back pain, as they may affect players’ mobility and ability. Despite the importance of this topic, no systematic reviews of back pain and postural changes in basketball players have been published.

Therefore, this systematic review aimed to evaluate the prevalence of back pain whose causes remain uncertain and probably multifactorial and musculoskeletal disorders in basketball players. This research is intended to identify factors that are associated with musculoskeletal disorders and pain in basketball players to contribute to the implementation of spinal pain and injury prevention and treatment interventions and programs to improve the health, quality of life, and sports performance of basketball players.

## 2. Materials and Methods

### 2.1. Protocol and Registration

The protocol (PROSPERO CRD42020201653) of this systematic review was registered in the International Prospective Register of Systematic Reviews and was published as an article [17]. The systematic review followed the Preferred Reporting Items for Systematic Reviews and Meta-Analyses methodology [18]. Ethical approval was not required because this review involved an analysis of previously published data.

### 2.2. Identification of Relevant Studies

Studies published in English were searched without any restriction on the publication period. The search was performed in three databases (Embase, PubMed, and Scopus) on 6 March 2022. The main search terms were “back pain”, “postural changes”, “players”, and “basketball”. The general search strategy is described in Table 1 and was adapted for the different databases (Appendix A).

**Table 1 healthcare-11-01190-t001:** General search strategy for the identification of published articles.

1	(“musculoskeletal disorder” OR “musculoskeletal disorders” OR “musculoskeletal disease” OR “musculoskeletal diseases” OR “musculoskeletal injuries” OR “musculoskeletal injury” OR posture OR “postural evaluation” OR “postural changes” OR scoliosis OR kyphosis OR lordosis OR spondylolysis OR “back pain” OR “low back pain” OR “back injuries” OR “lumbar pain” OR “neck pain” OR “spinal pain” OR “abnormalities in spine” OR “spine pain” OR “cervical pain” OR backache OR backaches OR “back ache” OR “back aches” OR “cumulative trauma disorders”)
2	(players OR player OR sportsman OR athletes OR athlete OR sportsmen OR sportswoman OR sportswomen)
3	(basket OR basketball OR sports OR sport)
4	(1) AND (2) AND (3)

The review followed the PECO structure (population, exposure, comparators, and outcomes) [19]. In this article, “P” represents players, “E” represents basketball sport, “C” represents spine regions (lumbar spine, cervical spine, thoracic spine), and “O” represents the prevalence of back pain and musculoskeletal disorders (postural changes and spinal injuries) in basketball players and associated factors.

Musculoskeletal disorders include fractures, acute soft tissue injuries (i.e., bruises, sprains, or strains); non-articular and non-rheumatic soft tissue disorders, tissue disorders including local myofascial pain syndrome and systemic fibromyalgia, arthritis, neurological disorders, amputations, and problems of postoperative rehabilitation following interventional orthopedic procedures [20]. Functional disorders that were evaluated in this review included traumatic injuries of the spine and injuries of the spinal cord, nerve roots, bone structure, and disk ligaments of the spine [21]. Postural changes considered included scoliosis, kyphosis, and lordosis. Back pain was defined as pain in the cervical, thoracic, and/or lumbar spine [22,23,24].

Inclusion criteria were: (a) basketball players of both sexes; (b) age up to 50 years (articles with other age groups were included, provided the data were presented separately); (c) observational studies (longitudinal, cross-sectional, cohort, and case-control); (d) assessment of musculoskeletal disorders or back pain in basketball players (articles with other injuries were included, provided the data are presented separately); (e) publications in English; and (f) studies with players from communities of different nationalities. Both specific and non-specific back pain were included.

Articles based on the following criteria were excluded: (a) studies with basketball players that pertained to injuries in other body regions (knee, shoulder, hip, ankle) unless back and spine data were presented separately or could be calculated; (b) paralympic athletes and/or players with physical or mental disabilities; (c) mixed sports samples, unless basketball player data were presented separately or could be calculated; (d) experimental studies; and (e) studies with incomplete data.

This search included full articles published without the restriction of the search period, and excluded books, book chapters, case reports, commentaries, letters, editorials, and systematic reviews. Articles for which the full text could not be retrieved from online databases were requested by email from the authors of the papers.

### 2.3. Study Selection

Articles found in the databases were imported into Mendeley [25] software, where we excluded duplicate studies. Subsequently, the Rayyan software [26] was used to read the titles and abstracts of the studies, and articles that did not meet the previously established eligibility criteria were excluded. Next, the full texts of the selected studies were read to confirm their eligibility. All steps were performed by two reviewers (SCCB and MSVF), and disagreements, if any, were resolved by a third reviewer (MN). The flowchart of the study selection is shown in Figure 1.

### 2.4. Data Extraction

The following data were extracted from the selected studies: author and year of publication, type of study, country of origin, population, sex, age group, type of change, tool, prevalence of postural change and injuries, location, and prevalence of back pain. The full description of the extracted data is included in Table 2, Table 3, Table 4 and Table 5.

We performed a meta-analysis with the data that were extracted to ascertain the prevalence of back pain, postural changes, and spinal injuries. However, these data were insufficient to perform a meta-analysis of the associated factors.

### 2.5. Examiner Training

Authors participating in the eligibility assessments were trained in the inclusion/exclusion criteria for studies and assessed the eligibility of 50 sample abstracts before they began reviewing the articles [27]. Besides performing standardized analyses using Mendeley and Rayyan software [25,26], the authors were trained to use risk-of-bias analysis tools through the examination of five non-included articles.

### 2.6. Methodological Quality and Risk of Bias

The included articles were assessed for methodological quality and risk of bias by using the Grading of Recommendations, Assessment, Development, and Evaluations-GRADE [28] or the Downs and Black checklist [29]. GRADE is a method that serves transparency and simplicity by enabling the classification of the quality of evidence into four categories: very low quality, low quality, moderate quality, and high quality. GRADE has been adopted worldwide because of its rigorous methodological classification and ease of use [28].

An adapted version of the Downs and Black checklist that was proposed by Noll was used [30], wherein each item indicates: (A) clearly stated objective; (B) clearly described main outcomes; (C) clearly defined sample characteristics; (D) clearly described distribution of main confounders; (E) clearly defined main findings; (F) random variability in estimates provided; (G) loss to follow-up described; (H) probability values provided; (I) representative target sample of the population; (J) representative sample recruitment of the population; (K) analyses adjusted for different follow-up times; (L) properly used statistical tests; (M) valid/reliable primary outcomes; (N) sample recruited from the same population; (O) adequate adjustment for confounders; and P) sample loss to follow-up considered (corresponding to items 1–3, 5–7, 9–12, 17, 18, 20, 21, 25, and 26). Items G and P were applied only to longitudinal studies, whereas items K and N were applied only to the case-control and longitudinal studies. Scores reach 100% at 12, 14, and 16 points for cross-sectional, case-control, and longitudinal studies, respectively. Scores above 70% were used to define a low risk of bias [29]. The Downs and Black checklist was identified as one of the two tools that were most frequently used in systematic reviews, which were registered in PROSPERO from 2011 to 2018 [31].

The scores were used to determine the methodological quality of the studies while considering five aspects: presentation, external validity, internal validity-bias, internal validity-confounders, and statistical power for inferences [29]. The risk of bias was assessed independently by two examiners. The prevalence of data identified in the studies was used for the data synthesis strategy.

### 2.7. Statistical Analysis

Meta-analyses were performed to estimate the prevalence of back pain, spinal injuries, and postural changes. Data were presented graphically in Forest plots to estimate the prevalence rates with 95% confidence intervals (CI). Statistical I² values were calculated to quantify the degree of heterogeneity between studies, with values of 25–50% representing moderate heterogeneity and values > 50% representing large heterogeneity among the studies [32]. Publication bias was assessed using Egger’s test. All analyses were performed using STATA (version 16.0; StataCorp, College Station, TX, USA).

## 3. Results

The stages of study selection are presented in Figure 1. According to the eligibility criteria, 6443 studies were identified, of which 1852 duplicate records were excluded. After screening the title and abstract, 4591 articles were selected; among these, 4506 did not meet the eligibility criteria, and 85 articles were included for a full-text review.

In this stage, 52 studies were excluded because they assessed other sports (n = 19), had different outcomes (n = 30), or were missing data (n = 3). Finally, 33 studies were included in this systematic review [6,7,8,10,11,16,33,34,35,36,37,38,39,40,41,42,43,44,45,46,47,48,49,50,51,52,53,54,55,56,57,58,59] (Table 2). Moderate interrater agreement (98%) was found between the two examiners.

### 3.1. Risk of Bias and Evaluation of the Quality of Studies

Of the articles included, 69.7% of the studies obtained ethical approval (n = 23) [6,7,10,11,16,34,35,36,38,39,40,44,45,46,47,48,50,51,53,54,55,56,59] and 51.5% clearly reported that there were no conflicts of interest (n = 17) [6,7,10,11,16,33,34,36,37,38,45,46,50,51,53,56,58].

The quality of evidence was low quality in 57.6% (n = 19) [6,10,11,16,37,38,39,40,41,42,43,44,45,46,47,48,49,50,51] and very low quality in 42.4% (n = 14) [7,8,33,34,35,36,52,53,54,55,56,57,58,59] of studies, respectively (Table 3). A total of 78.8% of the articles had a low risk of bias (n = 26) [6,7,8,10,11,16,33,34,36,37,38,39,40,41,42,43,44,45,46,47,49,50,51,54,57,58].

### 3.2. Main Characteristics of the Studies

In total, 72.7% (n = 24) of the articles were published between 2011 and 2022 [6,7,8,10,11,16,33,34,36,38,39,40,41,42,45,46,47,49,50,51,52,56,57,58]; 45.4% of the studies were conducted on the European continent (n = 15) (Table 4) [7,8,10,16,33,34,37,39,42,46,47,48,49,50,57]. With regard to the type of pain, back pain was addressed in 60.6% of the articles (n = 20) [6,7,8,11,34,35,37,38,42,43,46,47,49,50,54,55,56,58,59]; 48.5% of the articles investigated spine injuries (n = 16) [35,36,39,40,41,43,44,45,46,48,51,52,53,54,55,57].

The majority of the studies, 69.7% (n = 23), included participants from both sexes [6,8,10,33,34,37,38,40,41,42,43,44,45,46,47,48,49,50,52,53,54,55,58]. The average age of the participants ranged from 11.47 ± 2.10 [33] to 24.4 ± 4.7 years [36]. The sample size varied widely across studies, with a minimum of 10 [16] and a maximum of 5,566,124 players [41]. Furthermore, 63.6% (n = 21) of the articles were cross-sectional [6,7,8,11,34,37,38,40,41,42,44,45,47,48,52,53,54,55,57,58,59] (Table 4).

The vast majority of studies, a total of 14, did not present the period in which low back pain was assessed [6,38,39,40,41,43,44,45,46,48,52,55,57]. A total of 8 studies assessed low back pain in the last 3 to 6 months [33,35,36,37,42,51,56,59], 5 studies assessed it over a period of one year [10,16,34,47,50], 4 studies assessed it over a lifetime [7,8,11,54], and finally, 2 studies assessed it within the last 3 to 5 five years [49,53]. The detailed characteristics of the studies included in the systematic review are shown in Table 4.

**Table 2 healthcare-11-01190-t002:** Characteristics of the studies included in the systematic review and their outcome variables.

Author/Year	Design Country	No. Participants (% Male)	Age (Years/Mean and Standard Deviation)	Training Level	Tool	Postural Changes/Prevalence	Spine Injuries/Prevalence	Definition of Back Pain	Pain Location/Prevalence
Abdollahi et al. [51]	Retrospective Iran	204 (100%)	26.37 ± 7.42	Professional Super League and First Division League	Retrospective Injury Questionnaire (RIQ)	*	Upper back injury: 6.38% Lumbar injury: 48.53%	*	*
Auvinen et al. [37]	Cross-sectional Finland	4314 (51.5%)	15 to 16 years	Moderate to vigorous Up to once a month, 2–4 times a month, and at least twice a week	Questionnaire	*	*	*	All: 88.5%, up to once a month 24.6%, 2–4 times a month; 9.4%, at least twice a week Neck pain: 44.5%, up to once a month (reference group) 12.2%, 2–4 times a month 4.7%, at least twice a week Backache: 44.3%, up to once a month (reference group) 12.4%, 2–4 times a month −4.7%, at least twice a week.
Farahbakhsh et al. [7]	Cross-sectional Iran	52 (100%)	16.1± 1.1 years	Hours/week 11.6± 8.2	Questionnaire	*	*	Point prevalence, Prevalence of chronic pain Yearly prevalence, Sports-lifePrevalence, Lifetime prevalence	Total: Point prevalence 61.6% (N = 32) Prevalence of chronic pain 28.8% Neck pain: 36.53% 13.46% 36.53% 46.15% 57.69%. Low back pain: 25% 15.38% 50% 65.38% 63.46%
Grabara [52]	Cross-sectional Poland	52 (57.7%)	14 to 17 years	Training experience [years] 4.05 ± 0.58	Rippstein plurimeter	Hyperkyphosis: 21.2% Hypolordosis: 42.3% Hyperlordosis: 13.4%	*	*	*
Grabara [16]	Longitudinal Follow-up: 2 years Poland	10 (100%)	13–15 years	Training over a 2-year period	Plurimeter-V Gravity Inclinometer	Hyperkyphosis: 70% in almost 3 months after engagement in regular sports activity 70% after 1 year 60% after 2 years Hypokyphosis: 0% in almost 3 months after engaging in regular sports activity 20% after 1 year 10% after 2 years Hyperlordosis: 0% in almost 3 months after engaging in regular sports activity 10% after 1 year 0% after 2 years Hypolordosis: 70% in nearly 3 months after engaging in regular sports activity 30% after 1 year 50% after 2 years	*	*	*
Greene et al. [53]	Cross-sectional Follow-up: 1 year United States	33 (57.6%)	19 ± 1 years	College athletes	16-item questionnaire	*	Low back injury: Injury during the 1999/2000 season 18.2% History of low back injury in the last 5 years 27.3%	*	*
Habelt et al. [39]	Longitudinal Follow-up: 10 years Switzerland	168 (100%)	10–19 years	*	Clinical examination, Radiographic assessment (anteroposterior and lateral view), ultrasound, or MRI scan.	*	Spine injury: 1.8%	*	*
Hagiwara et al. [38]	Cross-sectional Japan	590 (56.1%)	6–15 years	Training per day on weekends (hours) ≤3304 (51.5) and >3286 (48.5)	Self-reported questionnaires	*	*	*	Low back pain: 12.9%
Hangai et al. [54]	Cross-sectional Japan	63 (69.8%)	19.7 ± 0.9 years	Athletes’ career time (years) 9.2 ± 1.8	MRI and clinical examination/Self-reported questionnaire	*	Disk Degeneration: 42.9%	*	Low back pain: during life: 81% During the previous 4 weeks: 17.7%
Hickey et al. [59]	Cross-sectional Australia	49(0%)	16–18 years	Young elite women’s basketball players	Medical records	*	Disk-related pain, spondylolysis, and acute fracture of a lumbar transverse process: 14.9%	Mechanical/facet joint-related low back pain Acute-chronic	Low back pain: 6.3%
Ichikawa et al. [55]	Cross-sectional Japan	16 male and female athletes	*	*	Radiography	*	Spondylolysis: 12.5%	*	Low back pain: 25%
Iwamoto et al. [43]	Retrospective 14-year period Japan	1229 (55.2%)	11–49 years	Class 2 = low recreational: sports activity once or twice a week; Class 3 = high recreational: sports activity; >3 times/week, and belonging to an elementary or high school team or other sports team; Class 4 = competitive: competitive sports activity and belonging to a professional, semi-professional, or university sports team	Radiographies or MRI/Database	*	Lumbar disc disease: 6.6.% Lumbar Spondylolysis: 2%	Non-traumatic pain	Low back pain: 2.9%
Keene et al. [44]	Cross-sectional United States	216 (male and female athletes)	*	College athletes	Review of training room medical records and hospital files	*	Total: 5.6% Strain: 5.1% Sprain: 0.5%	*	*
Kerr et al. [45]	Cross-sectional United States	19,991 (61.5%)		Athlete exposure was defined as the participation of 1 athlete in 1 school-sanctioned training or competition	Online Injury Surveillance System: Reporting Information Online	*	Trunk Displacements/Separations: 12.9% M = 7.7% F = 5.2%	*	*
Leppänen et al. [46]	Prospective Study Finland	201 (49.7%)	15.7 ± 1.7 years	College athletes	Questionnaire including information such as age, sex, injury history, playing experience, and family history of musculoskeletal disorders	*	Muscle/tendon: 26.9% Joint/ligament: 4.5% Bone injury: 5.5%	*	Low back pain: 9.95%
Meron et al. [41]	Cross-sectional United States	5,566,124 (55.6%)	*	An athlete exposure was defined as an athlete (154) participating in one practice, competition, or performance	High School Reporting Information Online injury surveillance system	*	Cervical spine injury: 0.0006%	*	*
Nagano et al. [56]	Prospective study Japan	54 (0%)	19.0 ± 2.8 years	College athletes	Modified Japanese version of the OSTRC questionnaire	*	*	*	Backache: 14.4%
Noormohammadpour et al. [11]	Cross-sectional Iran	140 (0%)	22.7 ± 2.7 years	Female college athletes competing in the National College Student Sports Olympics	Self-reported questionnaire	*	*	Point prevalence Yearly prevalence Sports-life Prevalence Lifetime Prevalence	Low back pain: 22.9% 47.9% 48.6% 68.6%
Nowak et al. [57]	Cross-sectional Poland	58 (100%)	17 ± 1.4 years	Professional players (club players), amateur league (amateur players)	Original questionnaire consisting of 28 items	*	Neck injury and back injury: 12.1%	*	*
Owen et al. [58]	Cross-sectional Japan	63 male and female athletes	20 ± 1 years	Well-trained male and female athletes who have spent a minimum of 5 years playing the sport	Subjective questionnaire	*	*	*	Back pain: 1.6%
Pasanen et al. [47]	Cross-sectional Finland	207 (48.8%)	14.9 ± 1.6 years	Young players that were official members of participating teams and had played official games in the previous season.	Questionnaire based on the Nordic standardized musculoskeletal symptoms questionnaire and on its modified version for athletes.	*	*	*	Low Back Pain: 45.4%
Rossi et al. [48]	Cross-sectional A retrospective analysis Italy	174 male and female athletes	15–27 years	Athletes referred to the Institute of Sport Sciences of the Italian Olympic Committee	Radiographic findings	*	Spondylolysis with low back pain: 9.77%	*	*
Rossi et al. [49]	Longitudinal 3-year follow-up Finland	203 (49.3%)	14.9 ± 1.6 years	Training hours (mean, standard deviation): 215.1 (102.9)	Nordic standardized questionnaire on musculoskeletal symptoms/modified version for athletes	*	*	Non-traumatic Acute Traumatic / AcuteTraumatic	Total: 11.9% Low back pain: 8.4% 3% Back pain: 0.5%
Rossi et al. [10]	Longitudinal 3-year follow-up Finland	271 male and female athletes	16.2 ± 1.7 years	Hours of team practice during follow-up, average hours: 244.8	Nordic standardized questionnaire on musculoskeletal symptoms/modified version for athletes	*	*	*	Low back pain: 46%
Rossi et al. [50]	Prospective cohort study Finland	128 male and female athletes	14.7 ± 1.5 years	Elite basketball players	Nordic standardized questionnaire of musculoskeletal symptoms	*	*	*	Low back pain: 25%
Sarcevic and Tepavcevic [33]	Case-control Serbia	38 male and female athletes Group of cases: 19 Control group: 19	11.5 ± 2.1 years 11.7 ± 1.9 years	Physical activity level, hours per week 3.03 ± 0.55, 3.04 ± 0.64	Checkup-Adams’ forward Bend test and scoliometer measurement	Adolescent idiopathic scoliosis: 100%	*	*	*
Schneider et al. [42]	Cross-sectional Germany	182 (70.9%)	15.5 ± 1.3 years	Elite youth basketball players from Germany’s three elite youth leagues	Sets of items from a previously validated and tested questionnaire	*	*	*	Back pain: 7 days: 34.3% 12-month prevalence rates: 70.9% More intense pain: 16.4% Neck pain: 7 days: 26.5% 12-month prevalence rates: 65.2% More intense pain: 6.1%
Schulz et al. [34]	Cross-sectional Germany	11 (18.2%)	19 years old	Basketball played 20.6 h per week	Self-developed survey with 59 items	*	*	*	Back pain: 54.5%
Selhorst et al. [40]	Cross-sectional United States	194 (60.3%)	15.0 ± 1.8 years	*	Radiographies	*	Spondylolysis: 33%	*	*
Silva et al. [35]	Longitudinal Brazil	66 (0%)	23 years	Elite Women’s Basketball Athletes. The teams played on average twice a week, and trained on average five times a week, which resulted in 76 matches and 375 training sessions.	Injury data were recorded by a physical therapist	*	Low back/back/neck injury: 33.3%	*	Low back pain: 12.1%
Trompeter et al. [8]	Cross-sectional Germany	518 (46.5%)	20.9 ± 4.8 years	Elite German athletes participating in the German Confederation of Olympic Sports	The questionnaire was based on the Nordic Questionnaire and a questionnaire developed by von Korff	*	*	Lifetime prevalence; 12-month prevalence; Point prevalence	Back pain: 91% 91% 67% Low back pain: 91% 86% 43%
Weiss et al. [36]	Prospective cohort study 24-week follow-up New Zealand	13 (100%)	24.4 ± 4.7 years	Competitive experience, 5.9 ± 3.6 years	Self-reported OSTRC injury questionnaire	*	Excessive use of lower back: 15.4%	*	*
Yabe et al. [6]	Cross-sectional Japan	592 (56.1%)	12–14 years	Training per day during the week: 2 h on average	Self-reported questionnaire	*	*	*	Low back pain: 12.8%

* Information missing in the article; ±, Standard deviation; BP, Back pain; F, Female; M, Male.

**Table 3 healthcare-11-01190-t003:** Assessment of methodological quality and strength of evidence.

Study (Year)	Conflicts ofInterest	Ethical Approval	Downs and Black Checklist	GRADE
A	B	C	D	E	F	G	H	I	J	K	L	M	N	O	P	Total	Score ^#^
Abdollahi et al. [51]	No	*	1	1	1	1	1	1	-	1	1	1	-	1	1	-	1	-	12/12	100%	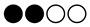
Auvinen et al. [37]	No	Yes	1	1	1	1	1	1	-	1	1	1	-	1	1	-	1	-	12/12	100%	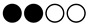
Farahbakhsh et al. [7]	No	Yes	1	1	1	0	1	1	-	1	0	0	-	1	1	-	1	-	9/12	75%	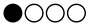
Grabara [52]	*	Yes	1	1	1	0	1	1	-	0	0	0	-	1	1	-	0	-	7/12	58.3%	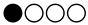
Grabara [16]	No	Yes	1	1	1	0	1	1	1	1	0	0	1	1	1	1	0	1	12/16	75%	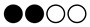
Greene et al. [53]	*	Yes	1	1	1	0	1	0	-	1	0	0	-	1	1	-	0	-	7/12	58.3%	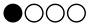
Habelt et al. [39]	*	*	1	1	1	0	1	1	1	0	1	0	1	1	1	1	0	1	12/16	75%	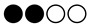
Hagiwara et al. [38]	No	Yes	1	1	1	1	1	1	-	1	1	1	-	1	1	-	1	-	12/12	100%	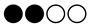
Hangai et al. [54]	No	Yes	1	1	1	1	1	1	-	1	0	1	-	1	1	-	1	-	11/12	91.7%	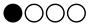
Hickey et al. [59]	*	Yes	1	1	1	0	1	0	-	0	1	1	-	1	1	-	0	-	8/12	66.7%	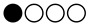
Ichikawa et al. [55]	*	*	1	1	0	0	1	1	-	0	0	1	-	1	1	-	0	-	7/12	58.3%	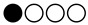
Iwamoto [43]	*	*	1	1	1	1	1	1	-	1	1	1	-	1	1	-	0	-	11/12	91.7%	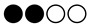
Keene et al. [44]	*	*	1	1	1	0	1	1	1	1	0	1	1	1	1	1	0	1	13/16	81.3%	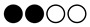
Kerr et al. [45]	No	Yes	1	1	1	0	1	1	-	1	1	1	-	1	1	-	0	-	10/12	83.3%	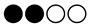
Leppänen et al. [46]	No	Yes	1	1	1	0	1	0	-	1	1	1	-	1	1	-	1	-	10/12	83.3%	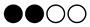
Meron et al. [41]	*	Yes	1	1	1	0	1	1	-	1	1	1	-	1	1	-	0	-	10/12	83.3%	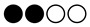
Nagano et al. [56]	No	Yes	1	1	1	0	1	0	-	1	0	0	-	1	1	-	1	-	8/12	66.7%	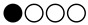
Noormohammadpour et al. [11]	No	Yes	1	1	1	1	1	1	-	1	1	1	-	1	1	-	1	-	12/12	100%	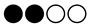
Nowak et al. [57]	*	*	1	1	1	0	1	1	-	1	0	1	-	1	1	-	0	-	9/12	75%	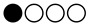
Owen et al. [58]	No	Yes	1	1	1	1	1	1	-	1	1	1	-	1	1	-	1	-	12/12	100%	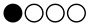
Pasanen et al. [47]	*	Yes	1	1	1	0	1	1	-	1	1	1	-	1	1	-	1	-	11/12	91.7%	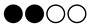
Rossi et al. [48]	*	*	1	1	1	0	1	0	-	0	1	1	-	1	1	-	0	-	8/12	66.7%	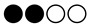
Rossi et al. [49]	*	Yes	1	1	1	1	1	1	-	1	1	1	-	1	1	-	1	-	12/12	100%	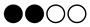
Rossi et al. [10]	No	Yes	1	1	1	1	1	1	1	1	1	1	1	1	1	1	1	0	15/16	93.8%	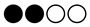
Rossi et al. [50]	No	Yes	1	1	1	0	1	1	-	1	1	1	-	1	1	-	1	-	11/12	91.7%	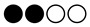
Sarcevic et al. [33]	No	Yes	1	1	1	1	1	1	-	1	0	1	1	1	1	1	0	-	12/14	85.7%	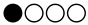
Schneider et al. [42]	*	Yes	1	1	1	0	1	1	-	1	1	1	-	1	1	-	0	-	10/12	83.3%	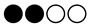
Schulz et al. [34]	No	*	1	1	1	0	1	1	-	1	1	0	-	1	1	-	0	-	9/12	75%	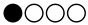
Selhorst et al. [40]	*	*	1	1	1	0	1	0	-	1	1	1	-	1	1	-	0	-	9/12	75%	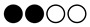
Silva et al. [35]	*	Yes	1	1	1	0	1	0	-	1	0	1	-	1	1	-	0	-	8/12	66.7%	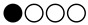
Trompeter et al. [8]	*	*	1	1	0	0	1	1	-	1	1	0	1	1	1	1	0	-	10/14	71.4%	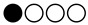
Weiss et al. [36]	No	Yes	1	1	1	0	1	1	-	1	0	1	1	1	1	1	0	-	11/14	78.6%	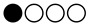
Yabe et al. [6]	No	Yes	1	1	1	1	1	1	-	1	1	0	-	1	1	-	1	-	11/12	91.7%	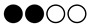

Downs and Black checklist: (A) clearly stated objective; (B) clearly described main outcomes; (C) clearly defined sample characteristics; (D) clearly described distribution of main confounders; (E) clearly defined main findings; (F) random variability in the estimates provided; (G) loss to follow-up described; (H) probability values provided; (I) representative target sample of the population; (J) representative sample recruitment of the population; (K) analyses adjusted for different follow-up times; (L) properly used statistical tests; (M) valid/reliable primary outcomes; (N) sample recruited from the same population; (O) adequate adjustment for confounders; and (P) sample loss to follow-up considered (corresponding to items 1–3, 5–7, 9–12, 17,18, 20, 21, 25, 26). *, not informed, -, not applied. Items G and P were applied to only longitudinal studies. Items K and N were applied to only case-control and longitudinal studies. # Scores reach 100% at 12, 14, and 16 points for cross-sectional, case-control, and longitudinal studies, respectively. GRADE, Grading of Recommendations, Assessment, Development, and Evaluations; were one filled circle, very low quality; two filled circles, low quality; three filled circles, moderate quality; and four filled circles, high quality.

**Table 4 healthcare-11-01190-t004:** Characteristics of the studies included in the systematic review.

Characteristics	Categories	Number of Studies (%)
Year of Publication	1982–2000	3 (9.0%)
	2001–2010	6 (18.2%)
	2011–2022	24 (72.7%)
Region		
America	Brazil USA	1 (3.1%) 5 (15.2%)
Africa	-	-
Asia	Iran Japan	3 (9%) 7 (21.1%)
Europe	Germany	3 (9%)
	Poland	3 (9%)
	Finland	6 (18.1%)
	Switzerland	1 (3.1%)
	Italy	1 (3.1%)
	Serbia	1 (3.1%)
Oceania	Australia New Zealand	1 (3.1%) 1 (3.1%)
Study design	Case-control	1 (3.1%)
	Retrospective	2 (6%)
	Prospective	4 (12.1%)
	Longitudinal	5 (15.2%)
	Cross-sectional	21 (63.6%)
Sex	Male only	6 (18.2%)
	Female only	4 (12.1%)
	Both sexes	23 (69.7%)
Sample size	<100	15 (45.5%)
	100–500	12 (36.4%)
	501–1000	2 (6%)
	>1000	4 (12.1%)
Participants	Postural changes	3 (9.1%)
	Back pain and spine injuries	6 (18.2%)
	Spine injuries	10 (30.3%)
	Back pain	14 (42.4%)

**Table 5 healthcare-11-01190-t005:** Associated Factors.

Author	Associated Factors
Auvinen et al. [37]	-
Farahbakhsh et al. [7]	The highest risk of neck pain at all times was observed among basketball players compared to other sports groups (*p* < 0.05; OR [95% CI 1.54–7.25]).
Grabara [52]	-
Grabara [16]	-
Greene et al. [53]	-
Habelt et al. [39]	-
Hagiwara et al. [38]	Upper limb pain was significantly associated with low back pain (OR: 7.86 [95% CI 3.93–15.72], *p* < 0.001). Shoulder pain was significantly associated with training per week (>4 days) (OR: 4.15; 95% CI: 1.29–13.40) and low back pain (OR: 13.77 [95% CI 5.70–33.24], *p* < 0.001).
Hangai et al. [54]	Logistic regression analysis of participants with disc degeneration, including a basketball group, adjusted for sex and obesity (OR: 1.61 [95% CI 0.78–3.35], *p* = 0.1982)
Hickey et al. [59]	-
Ichikawa et al. [55]	-
Iwamoto et al. [43]	-
Keene et al. [44]	-
Kerr et al. [45]	-
Leppãnen et al. [46]	Female players had a higher incidence of overuse injuries compared to male basketball players (IRR 1.61 [95% CI 1.07–2.46], *p* < 0.05. Previous injury was significantly associated with low-back overuse injuries in basketball and floorball players (OR 3.99 [CI 1.48–10.78], *p* = 0.01)
Meron et al. [41]	For sports that allow comparison between the sexes, females had higher basketball injury rates (RR, 2.02 [CI 1.01–4.03], *p* < 0.05)
Nagano et al. [56]	-
Noormohammadpour et al. [11]	-
Nowak et al. [57]	The differences in stretching before a workout or game between players training up to three times a week and players training four or more times a week were statistically significant (χ2 = 8.926, *p* = 0.012, V = 0.392)
Owen et al. [58]	After matching participants based on the status of back pain and height, basketball players showed signs of intervertebral disc hypertrophy (*p* ≤ 0.043)
Pasanen et al. [47]	Family history of musculoskeletal disorders (OR 2.02 [95% CI 1.22–3.34]) and higher age (OR 1.22 [95% CI 1.05–1.41]) were associated with low back pain in basketball and floorball players.
Rossi and Dragoni [48]	-
Rossi et al. [49]	-
Rossi et al. [10]	There was a small increase in the risk of low back pain with a one-degree decrease in the right leg during the SLVDJ landing (HR 1.09 [95% CI 1.02–1.17] per one-degree decrease in the APF). Basketball and floorball players. All LBP Femur–pelvic angle, right side HR 1.09 (1.02–1.17) 0.014 Gradual onset non-traumatic LBP Femur–pelvic angle, right side HR 1.09 (1.01 to 1.18) 0.021
Rossi et al. [50]	None of the risk factors investigated were associated with low back pain in univariate Cox analyses.
Sarcevic and Tepavcevic [33]	-
Schneider et al. [42]	-
Schulz et al. [34]	-
Selhorst et al. [40]	Presence of spondylolysis in male basketball athletes RR (95% CI) = 1.05 (0.89–1.24) Presence of spondylolysis in female basketball athletes RR (95% CI) = 0.98 (0.86–1.12) Overall: Male athletes were 1.5 times more likely to have spondylolysis than female athletes (*p* = 0.01).
Silva et al. [35]	Older athletes were more likely to have consecutive injuries than younger athletes during the study period. This comparison was statistically significant (*p* = 0.010).
Trompeter et al. [8]	Among basketball players, these problems, along with a high frequency of jumping and landing, can lead to back pain. Compared with control subjects, significantly higher rates of back pain were found in those who participated in elite rowing, dancing, fencing, gymnastics, underwater rugby, water polo, shooting, basketball, field hockey, track and field, ice hockey, and figure skating.
Weiss et al. [36]	The mean weekly prevalence of all reported overuse conditions was 63% (95% CI 60–66), and that of severe overuse conditions was 7.3% (95% CI: 7.1–7.6).
Yabe et al. [6]	Participants with lower extremity pain had higher rates of low back pain, with an OR (95% CI) of 6.21 (3.57–10.80), than participants without lower extremity pain. Moreover, there was a significant association between knee/ankle pain and low back pain. Compared with participants without knee/ankle pain, the OR (95% CI) for low back pain was 4.25 (2.55–7.07) for participants with knee pain and 3.79 (2.26–6.36) for participants with ankle pain.

CI, confidence interval; OR, odds ratio; RR, Relative Risk; Cramér’s V, V is a measure of association between two nominal variables; χ2, chi-square. SLVDJ: single leg drops vertical jump.

### 3.3. Assessment of the Prevalence of Spinal Injuries and Postural Changes

Of the tools used to assess spinal injuries, 31.3% were radiographs (n = 5) [39,40,43,48,55], 31.3% were questionnaires (n = 5) [36,46,51,53,57], and 18.8% comprised medical records (n = 3) [35,44,59]. The other studies used different tools to assess the prevalence of spine injuries, with 18.8% using magnetic resonance imaging (n = 3) (Table 3) [39,43,54]. Plurimeters (inclinometers) (n = 2) [16,52] were used in 66.7% of cases, and the Checkup-Adams’ forward bend test and scoliometer measurement (n = 1) [33] were used in 33.3% of cases to assess postural changes.

The most common diagnoses were spondylolysis in 31.3% of the studies (n = 5) [40,43,48,55,59], lumbar spine injuries in 25% (n = 4) [35,36,51,53], and cervical spine injuries in 18.8% (n = 3) [35,41,57]. Of these studies, 43.8% (n = 7) examined more than one postural change [35,43,44,46,51,57,59]. Among the postural changes, the most common abnormalities were hyperlordosis, hypolordosis, and hyperkyphosis in 66.7% of the studies (n = 2) [16,52].

For spinal injuries, the prevalence of low back injuries was 48.5% (n = 204) [51], whereas that of disk degeneration, spondylolysis, back injuries (lumbar, dorsal, and cervical), overuse injuries, and trunk displacement/separation in players was 42.9% (n = 63) [54], 33% (n = 194) [40], 33.3% (n = 66) [35], 15.4% (n = 13) [36], and 12.9% (n = 19,991), respectively [45]. With regard to postural changes, the prevalence of hyperkyphosis was 70% (n = 10) [16], and hypolordosis was 42.3% (n = 52) [52].

### 3.4. Assessment of the Prevalence of Back Pain

To assess the prevalence of back pain, self-reported questionnaires were used in 80% of the studies (n = 16) [6,7,8,10,11,34,37,38,42,46,47,49,50,54,56,58] and medical records in 10% (n = 2) [35,59]. In 80% (n = 16) of the studies, low back pain was identified [6,7,8,10,11,37,38,43,46,47,49,50,54,55,56,59]; in 25% (n = 5) of the studies reported back pain [8,34,42,49,58]. In 25% (n = 5) of the studies, more than one outcome was reported for the location of back pain [7,8,37,42,49].

The prevalence of low back pain and back pain was 91% in 21 players [8], the prevalence of low back pain was 81% in 63 players [54], and of back pain was 70.9% in 182 participants [42]. The prevalence of neck pain ranged from 44.5% in a sample of 4314 players [37] to 26.5% in a sample of 182 players [42].

### 3.5. Associated Factors

Studies reported sex and age as factors that were associated with musculoskeletal disorders. Female basketball players had a higher injury rate than male players (IRR, 1.11 [CI: 0.44–2.71]; RR, 2.02 [CI: 1.01–4.03]) [46,51]. Furthermore, higher age was associated with low back pain (OR, 1.22 [CI: 1.05–1.41], *p* < 0.008) [7] and were more likely to suffer consecutive injuries (*p* = 0.010) [40]. Data on the associated factors found in the articles are presented in Table 5 but were insufficient to perform a meta-analysis.

## 4. Meta-Analysis

The overall prevalence of back pain was 43% [confidence interval (CI) of 95%: −1% to 88%] (Figure 2). Statistical heterogeneity between studies was high (I2 = 91.76%, *p* < 0.001). Thus, we performed a meta-regression analysis (tau2 = 0, I2 = 0.00). The analysis showed that heterogeneity had no influence on the result of the analysis. Using Egger’s regression test, we found no evidence of publication bias in the meta-analysis of the overall prevalence of pain (*p* = 0.081).

The prevalence of neck pain was 36% [95% CI, 22–50%], back pain was 16% [95% CI, 4–28%] (Figure 3), of low back pain was 26% [95% CI, 16–37%], and of thoracic spine pain was 6% [95% CI, 3–9%]. The combined prevalence of pain was 26% [95% CI, 17–34%]. There was high statistical heterogeneity for both back pain (I2 = 97.04%, *p* = 0.001) and low back pain (I2 = 99.37%, *p* = 0.001). Similarly, we performed a meta-regression analysis (Tau2 = 20.82, I2 = 0.001). The analysis showed that heterogeneity had no influence on the outcome of the analysis. Using the Egger regression test, we found evidence of publication bias in the meta-analysis of pooled prevalence (*p* = 0.001).

The pooled prevalence of spine injury and spondylolysis was 10% [95% CI, 4–15%] (Figure 4). The prevalence of spine injury was 3% [95% CI, 1–5%], and spondylolysis was 14% [95% CI, 0.1–27%]. There was high statistical heterogeneity for spondylolysis (I2 = 96.85%, *p* = 0.001). Therefore, we performed a meta-regression (tau2 = 0, I2 = 0.001). The analysis showed that heterogeneity had no influence on the outcome of the analysis. Egger’s regression test showed no evidence of publication bias (*p* = 0.187).

The pooled prevalence of hyperkyphosis and hyperlordosis was 30% [95% CI, 9–51%] (Figure 5). The prevalence of hyperkyphosis was 28%, and of hyperlordosis was 13%. There was no evidence of statistical heterogeneity (I2 = 0.001).

## 5. Discussion

This is the first systematic review to examine the prevalence of back pain and musculoskeletal disorders in basketball players and to ascertain the associated factors. Our findings suggest a high overall prevalence of back pain, with neck pain and low back pain being the most prevalent. Among musculoskeletal disorders, spondylolysis was most prevalent among spinal injuries. The prevalence of hyperkyphosis was highest among postural changes. Sex and age were associated with musculoskeletal disorders in this review, but the data were insufficient to perform a meta-analysis.

This review found an overall prevalence of back pain of 43% in basketball players. These results corroborate the findings of Pasanen et al. [47], who showed that one in six young elite basketball players reported back pain as the predominant pain [47]. Our data indicate a prevalence of neck pain of 36% in basketball, which is consistent with Safiri et al. [60], which showed that the number of cases of neck pain in women was 166.0 million (118.7–224.8), whereas in men it was 122.7 million (87.1–167.5) [59].

Similarly, there were 568.4 million (95% IU: 505.0–640.6 million) cases of low back pain worldwide in 2019 [60], and our study showed a prevalence of 25% of low back pain in basketball players. According to Kim et al. [61], who studied college basketball players, the prevalence of low back pain was 69.8% in the last year of training and 84.1% throughout life [61]. In 2022, low back pain remained the largest contributor to the total number of cases of musculoskeletal disorders. There are 570 million cases worldwide, which account for 7.4% of years lived with disability [62].

Our study found a prevalence of spinal injuries of 3%. In addition to the direct effects of back injury on general health, the indirect effects of this injury could lead to the irreversible loss of future young athletes [1,2,3]. Furthermore, our study found a pooled prevalence of spine injuries and spondylolysis of 10% and a prevalence of spondylolysis of 14%. Spondylolysis is an anatomic defect or fracture of the pars interarticularis (part of the neural arch located between the superior and inferior articular facets) of the vertebral arch, which occurs in the fifth lumbar vertebra (L5) in 85% to 95% of cases [63]. The higher percentage of spondylolysis can be explained by the risk factors for the development of this injury, which includes repetitive hyperextension and rotation of the lumbar spine that may occur in sports such as basketball [64].

Given the strong association between low back pain and sports activities that involve hyperextension with rotation of the lumbar spine, spondylolysis is a major concern in adolescent athletes [65]. This suggests that more frequent basketball training may have a negative effect on spinal positioning in basketball players [66]. Our study showed a pooled prevalence of hyperkyphosis and hyperlordosis of 30%, with a prevalence of hyperkyphosis of 28% and of hyperlordosis of 13%, in the sample.

In the study of basketball players conducted by Nam et al. [67], the sample had curvature values in the range of 37–42°, which represents a difference of approximately 10° as compared with the normal range [68]. In addition, Kaplan [69], when comparing basketball players and a control group to determine posture, found that the basketball group had a lateral spinal curvature, head positioned in the right sagittal plane, pelvic tilt, and thoracic kyphosis [70].

The first limitation of this study is the limitation may be different periods of back pain; example, we have studies that did not mention the period in which the prevalence was verified [6,38,39,40,41,43,44,45,46,48,52,55,57], and we also have studies that verified the prevalence in the last three months [33,42] up to the last five years [53]. Second, lack of definition of back pain in many studies, and the fact that different tools were used in the assessment of back pain [69], spinal injuries, and postural changes in the included studies, which makes it difficult to compare the results. Third, some studies could not be fully retrieved. Fourth, most studies had a cross-sectional design, which does not allow an inference of cause and effect. The strengths of this study include the performance of a meta-analysis that provided a general estimate of the prevalence of back pain, spinal injuries, and postural changes in basketball players. To our knowledge, this is the first systematic review to summarize the evidence of the association between these musculoskeletal disorders in basketball players, which allows us to clarify some gaps in the literature and make recommendations for future research.

In this sense, future studies should consider the severity and duration of back pain, spine injuries, and postural changes to prevent players from withdrawing from sports for a long time [71]. The relationship between back pain, spine injuries, and postural changes should be further investigated, and the associated factors need to be analyzed. The results of our study indicate that clarifying the relationship between back pain, injuries, and postural changes is important for developing actions and programs to prevent and treat musculoskeletal disorders, thus contributing to the health and sports performance of basketball players. It is important for health professionals to be aware of the origin of back pain, spinal injuries, and postural changes, as well as the protective mechanisms that can be adopted for more effective interventions.

## 6. Conclusions

We found a significant overall prevalence of back pain in basketball players. On comparing the prevalence of this pain in the general population, the value found was lower, suggesting that basketball players have a lower prevalence of back pain than the general population. The most prevalent types of back pain were neck pain and low back pain, the most prevalent musculoskeletal disorder was spondylolysis, and the most prevalent postural change was hyperkyphosis.

## Figures and Tables

**Figure 1 healthcare-11-01190-f001:**
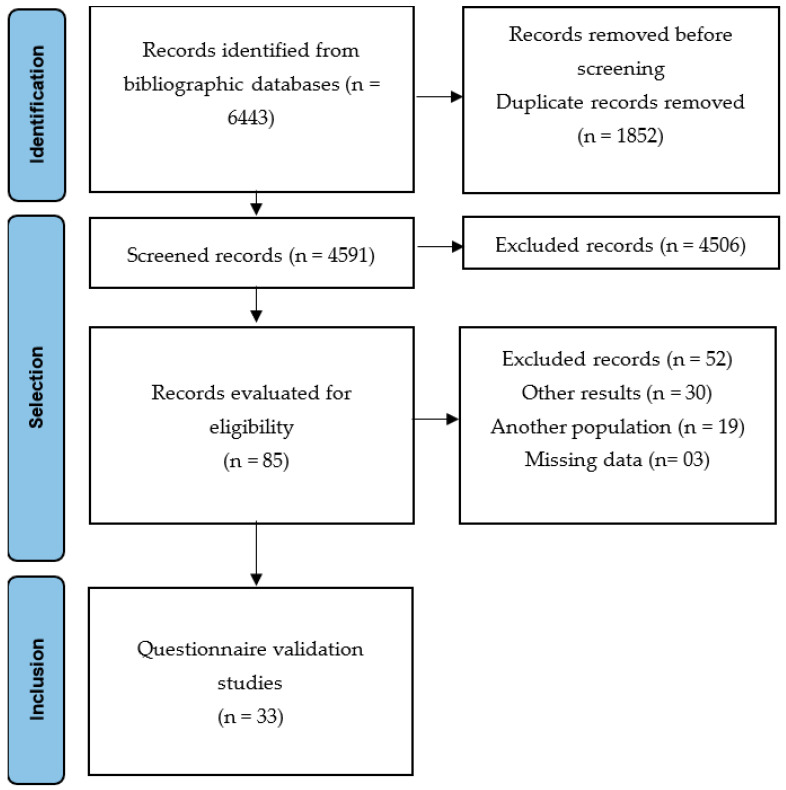
Preferred Reporting Items for Systematic Reviews and Meta-Analyses (PRISMA) flow diagram of articles included in this review.

**Figure 2 healthcare-11-01190-f002:**
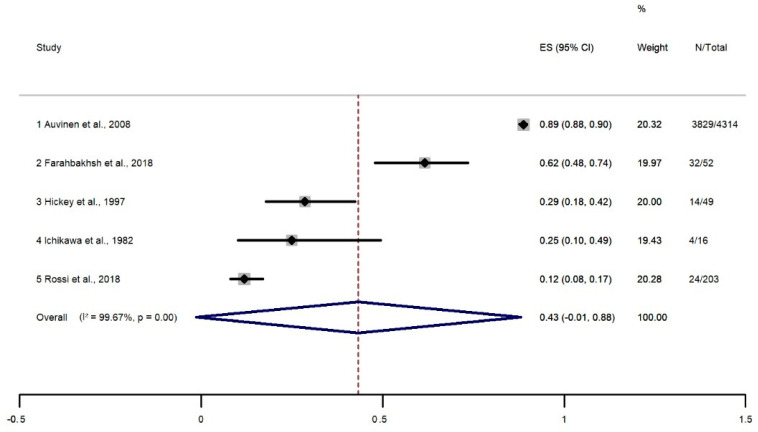
Meta-analysis of overall prevalence of back pain [7,10,37,55,59].

**Figure 3 healthcare-11-01190-f003:**
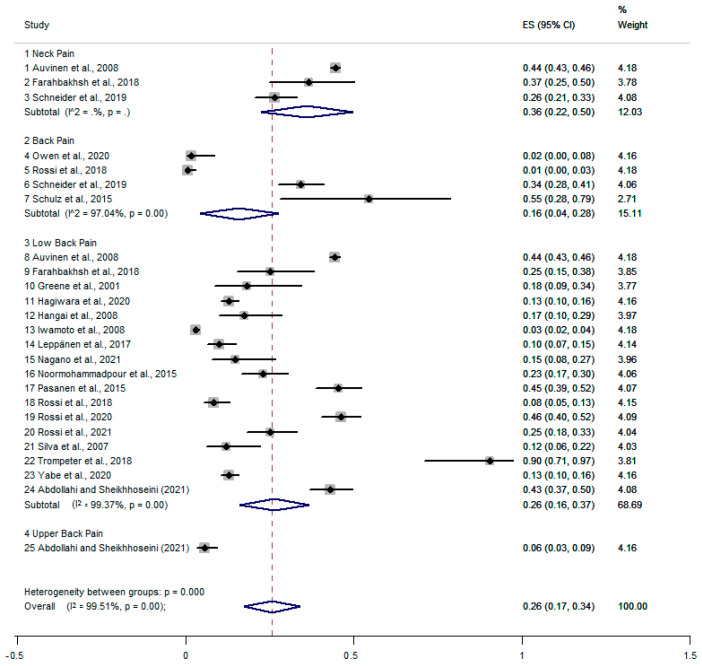
Meta-analysis of prevalence of neck pain, back pain, and low back pain [6,7,8,9,10,11,34,35,37,38,42,43,46,47,49,50,51,53,54,56,58].

**Figure 4 healthcare-11-01190-f004:**
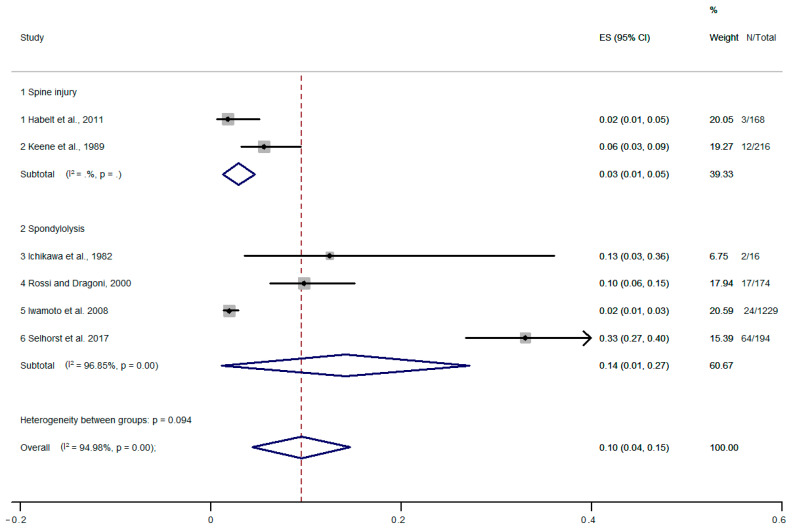
Meta-analysis of pooled prevalence of spinal injury and spondylolysis. [39,40,43,44,48,55].

**Figure 5 healthcare-11-01190-f005:**
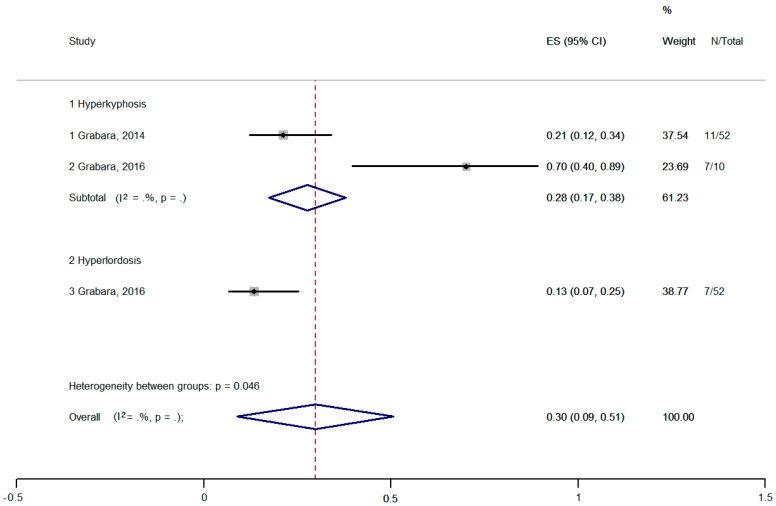
Meta-analysis of pooled prevalence of hyperkyphosis and hyperlordosis [16,52].

## Data Availability

Additional data can be obtained from the authors.

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
