# Peer review of "Prevalence and Risk Factors of Musculoskeletal Disorders in Basketball Players: Systematic Review and Meta-Analysis"

_healthcare, 2023, doi:10.3390/healthcare11081190_

Round 1

Reviewer 1 Report

Review Manuscript healthcare-2298321, entitled " Prevalence and risk factors of musculoskeletal disorders in basketball players: systematic review and meta-analysis."

Type of manuscript: Systematic Review

Journal: Healthcare

Dear Authors.

Back pain is the most common musculoskeletal disorders and a health problem in general population. Physical activity is significant for preventing and treating back pain (BP), and the increase in the level of physical activity has become an important part of recommendations in the management of BP. On the other hand, there is evidence that a high level of physical activity being hazardous on the lower back and some types of sports may be beneficial or harmful in developing or protecting against low back pain.

This study concerns an important problem of musculoskeletal disorders among athletes. This article aimed to evaluate the prevalence of back pain and musculoskeletal disorders in basketball players and is intended to identify factors that are associated with musculoskeletal disorders and pain in basketball players to contribute to the implementation of spinal pain and injury prevention and treatment interventions and programs to improve the health, quality of life, and sports performance of basketball players. However, I have a couple of major requests to be revised as stated below.

Comments:

1.     Lines 43-44: “Another study of 1114 elite German athletes from various sports reported an 89% prevalence of low back pain and a 77% prevalence of low back pain (8).” – I don`t understand this sentence? 89% or 77% - Why there are two values related to the prevalence of low back pain?

2.     How is back pain defined in this review? Is it specific or non-specific pain? Or maybe both?

3.     Figure 1 is missing (line 85).

4.     Prisma flowchart is missing too.

5.     Line 91: (5,8-10,15,32-58) - reference number 6 is missing

6.     In the Materials and Methods section, the criteria for inclusion and exclusion of articles should be added.

7.     There is no description of how to extract the data.

8.     I propose to change the citation in Tables 2 and 3: from Auvinien et al. (2008) to Auvinien et al. (40) - such as citations in the text

9.     References 16 to 31 are not included in the text.

10.  The numbering of articles included in this review should be changed: from 32-58 to 16-42.

11.  The description of the tool for methodology quality assessment of articles is missing. There is only a brief description under Table 3. There is also no literature related to this tool.

12.  Line 101: please check the citation order

13.  Lines 115-116: “Furthermore, 63.6% of the articles were cross-sectional (n=21)” - and what are the others? Such information should also be included in the inclusion criteria. What types of studies have been included in this review.

14.  Section 3.4: For how long did the subjects experience back pain? Lifetime prevalence, during last year, six months, three months or last week?

15.  Lines 152-154: “In 25% of the studies, more than one outcome (n=5) (6, 8,40,45,52) was reported for the location of back pain, and the most common locations were low back (in 153 80% of the studies; n=16) (5,6,8–10,32,33,40,41,46,49,50,52,53,58,59) and back (in 25% of 154 the studies; n=5) (8,35,37,45,52)”. The most common locations were low back and back – I do not understand. Did 5 articles refer to back pain without indicating a specific location of pain?

16.  Where is Table 5?

17.  Table 5 is not signed. Line 165: is it about Table 5?

18.  Table 6 is in a Supplementary Materials, line 271 – Table7. You need to correct the table number.

19.  Line 214: (87.1 million). at 167.5) (59) – You should correct it.

20.  Line 238: Nam et al. (2020) – You should correct it: Nam et al. (68)

21.  Line 240: Kaplan (2018) – Kaplan (69)

22.  The first limitation of this study is the fact that different tools were used in the assessment of back pain”. The limitation may be different periods of back pain and the lack of definition of back pain in many publications (see Table 2).

Thank you very much. I congratulate the authors for their work. I think the article should be corrected and supplemented with missing information.

Author Response

April 4, 2023.

Manuscript Number: Healthcare-2298321

Title: Prevalence and risk factors of musculoskeletal disorders in basketball players: systematic review and meta-analysis

Dear Editor,

We would like to thank you, the Editor, and the Reviewers for the thoughtful and in-depth comments in our manuscript. Your suggestions and remarks have helped us to reflect on our paper and improve it. We appreciate your commitment and effort. We have carefully considered every comment, promptly accepted all the suggestions, and made the alterations as recommended using the red color in the manuscript.

Please find below a point-by-point response to the Editors’ and Reviewers’ comments with answers in red font.

REVIEWER 1

Dear Authors.

Back pain is the most common musculoskeletal disorders and a health problem in general population. Physical activity is significant for preventing and treating back pain (BP), and the increase in the level of physical activity has become an important part of recommendations in the management of BP.

On the other hand, there is evidence that a high level of physical activity being hazardous on the lower back and some types of sports may be beneficial or harmful in developing or protecting against low back pain.

This study concerns an important problem of musculoskeletal disorders among athletes. This article aimed to evaluate the prevalence of back pain and musculoskeletal disorders in basketball players and is intended to identify factors that are associated with musculoskeletal disorders and pain in basketball players to contribute to the implementation of spinal pain and injury prevention and treatment interventions and programs to improve the health, quality of life, and sports performance of basketball players. However, I have a couple of major requests to be revised as stated below.

Comments:

1. Lines 43-44: “Another study of 1114 elite German athletes from various sports reported an 89% prevalence of low back pain and a 77% prevalence of low back pain (8).” – I don`t understand this sentence? 89% or 77% - Why there are two values related to the prevalence of low back pain?

Authors: Thanks for your careful review. Sorry for our typo. Based on your comment, we rewrote this sentences, as follows:

Line 44: “Another study of 1114 elite German athletes from various sports reported an 89% prevalence of low back pain [9].”

2. How is back pain defined in this review? Is it specific or nonspecific pain? Or maybe both?

Authors: Thanks for your observation. We work with nonspecific pain and to describe it better we include the following excerpt at the end of the introduction, as follows:

Lines 64 - 66: “…aimed to evaluate the prevalence of non-specific back pain whose causes remain un-certain and probably multifactorial and musculoskeletal disorders in bas-ketball players.”

3. Figure 1 is missing (line 85).

Authors: Thanks for your careful review. I believe that there was an error in the processing of our file, because in our original article there is Figure 1 which is the Preferred Reporting Items for Systematic Reviews and Meta-Analyses (PRISMA) flow diagram of articles included in this review, but only after your comment I noticed that figure 1 did not appear in the pdf made available to you. Therefore, we have provided for its inclusion again right after the second paragraph of the results section on page 4.

Below you can see the figure:

4. Prisma flowchart is missing too.

Authors: Thanks for your observation. Figure 1 is the PRISM Flowchart which is now included in the manuscript on page 4.

5. Line 91: (5,8-10,15,32-58) - reference number 6 is missing

Authors: Thanks for your careful review. All references have been corrected one by one.

6. In the Materials and Methods section, the criteria for inclusion and exclusion of articles should be added.

Authors: Thanks for your observation. Sorry for the mistake, I believe that when transferring the text to the template (three pages of the original text were lost) we only identified the lack of these three pages through your observation. The text that did not appear for your review starts right after Table 1 and ends before the results section. Especially in item 2.2 Identification of relevant studies are the inclusion and exclusion criteria, according to the paragraphs below:

Lines 101 – 113: “Inclusion criteria were: a) basketball players of both sexes; b) age up to 50 years (articles with other age groups were included, provided the data were presented separately); c) observational studies (longitudinal, cross-sectional, cohort, and case-control); d) assessment of musculoskeletal disorders and back pain in basketball players (articles with other injuries were included, provided the data are presented separately); e) publications in English; and f) studies with players from communities of different nationalities.

Articles based on the following criteria were excluded: a) studies with basketball players that pertained to injuries in other body regions (knee, shoulder, hip, ankle) unless back and spine data were presented separately or could be calculated; b) paralympic athletes and/or players with physical or mental disabilities; c) mixed sports samples, unless basketball player data were presented separately or could be calculated; d) experimental studies; and e) studies with incomplete data.”

7. There is no description of how to extract the data.

Authors: Thanks for your observation. Again, very sorry for the mistake, I believe that when transferring the text to the template (three pages of the original text were lost) we only identified the lack of these three pages through your observation. The text that did not appear for your review starts right after Table 1 and ends before the results section. Especially in item 2.4 Data Extraction, according to the paragraphs below:

Lines 151-157: “The following data were extracted from the selected studies: author and year of publication, type of study, country of origin, population, sex, age group, type of change, tool and prevalence of postural change and injuries, location and prevalence of back pain. The full description of the extracted data is included in Tables 2, 3, 4 and 5.

We performed a meta-analysis with the data that were extracted to ascertain the prevalence of back pain, postural changes, and spinal injuries. However, these data were insufficient to perform a meta-analysis of the associated factors.”

8. I propose to change the citation in Tables 2 and 3: from Auvinien et al. (2008) to Auvinien et al. (40) - such as citations in the text.

Authors: Thanks for your observation. We made the change as suggested in Tables 2 and 3.

9. References 16 to 31 are not included in the text.

Authors: Thanks for your careful review. All references have been corrected one by one.

10. The numbering of articles included in this review should be changed: from 32-58 to 16-42.

Authors: Thanks for your careful review. All references have been corrected one by one.

11. The description of the tool for methodology quality assessment of articles is missing. There is only a brief description under Table 3. There is also no literature related to this tool.

Authors: Thanks for your observation. Here we highlight the same mistake, I believe that when transferring the text to the template (three pages of the original text were lost) we only identified the lack of these three pages through your observation. The text that did not appear for your review starts right after Table 1 and ends before the results section. Especially in item 2.6 Methodological quality and risk of bias , according to the paragraphs below:

Lines 165-192: “The included articles were assessed for methodological quality and risk of bias by using the Grading of Recommendations, Assessment, Development, and Evaluations - GRADE [28] or the Downs and Black checklist [29]. GRADE is a method that serves transparency and simplicity by enabling the classification of the quality of evidence into four categories: very low quality, low quality, moderate quality, and high quality. GRADE has been adopted worldwide because of its rigorous methodological classification and ease of use [28].

An adapted version of the Downs and Black checklist that was proposed by Noll was used [30], wherein each item indicates: A) clearly stated objective; B) clearly described main outcomes; C) clearly defined sample characteristics; D) clearly described distribution of main confounders; E) clearly defined main findings; F) random variability in estimates provided; G) loss to follow-up described; H) probability values provided; I) representative target sample of the population; J) representative sample recruitment of the population; K) analyses adjusted for different follow-up times; L) properly used statistical tests; M) valid/reliable primary outcomes; N) sample recruited from the same population; O) adequate adjustment for confounders; and P) sample loss to follow-up considered (corresponding to items 1–3, 5–7, 9–12, 17, 18, 20, 21, 25, and 26). Items G and P were applied only to longitudinal studies, whereas items K and N were applied only to case–control and longitudinal studies. Scores reach 100% at 12, 14, and 16 points for cross-sectional, case–control, and longitudinal studies, respectively. Scores above 70% were used to define a low risk of bias [29]. The Downs and Black checklist was identified as one of the two tools that were most frequently used in systematic reviews which were registered in PROSPERO from 2011 to 2018 [31].

The scores were used to determine the methodological quality of the studies while considering five aspects: Presentation, external validity, internal validity - bias, internal validity - confounders, and statistical power for inferences [29]. Risk of bias was assessed independently by two examiners. The prevalence of data identified in the studies was used for the data synthesis strategy.”

12. Line 101: please check the citation order

Authors: Thanks for your careful review. All references have been corrected one by one.

13. Lines 115-116: “Furthermore, 63.6% of the articles were crosssectional (n=21)” - and what are the others? Such information should also be included in the inclusion criteria. What types of studies have been included in this review.

Authors: Thanks for your observation. Sorry for the mistake, I believe that when transferring the text to the template (three pages of the original text were lost) we only identified the lack of these three pages through your observation. The text that did not appear for your review starts right after Table 1 and ends before the results section. Especially in item 2.2 Identification of relevant studies are the inclusion and exclusion criteria, according to the paragraphs below:

Lines 101 – 113: “Inclusion criteria were: a) basketball players of both sexes; b) age up to 50 years (articles with other age groups were included, provided the data were presented separately); c) observational studies (longitudinal, cross-sectional, cohort, and case-control); d) assessment of musculoskeletal disorders and back pain in basketball players (articles with other injuries were included, provided the data are presented separately); e) publications in English; and f) studies with players from communities of different nationalities.

Articles based on the following criteria were excluded: a) studies with basketball players that pertained to injuries in other body regions (knee, shoulder, hip, ankle) unless back and spine data were presented separately or could be calculated; b) paralympic athletes and/or players with physical or mental disabilities; c) mixed sports samples,, unless basketball player data were presented separately or could be calculated; d) experimental studies; and e) studies with incomplete data.”

14. Section 3.4: For how long did the subjects experience back pain? Lifetime prevalence, during last year, six months, three months or last week?

Authors: Thanks for your careful review. Due to no limitation of this period, we added as suggested as a limitation of our study.

Lines: 384-385 “…limitation may be different periods of back pain, lack of definition of back pain in many studies…”

15. Lines 152-154: “In 25% of the studies, more than one outcome (n=5) (6, 8,40,45,52) was reported for the location of back pain, and the most common locations were low back (in 153 80% of the studies; n=16) (5,6,8–10,32,33,40,41,46,49,50,52,53,58,59) and back (in 25% of 154 the studies; n=5) (8,35,37,45,52)”. The most common locations were low back and back – I do not understand. Did 5 articles refer to back pain without indicating a specific location of pain?

Authors: Thanks for your observation. Exactly, 5 articles reported back pain without indicating the specific location.

16. Where is Table 5?

Authors: Thanks for your careful review. Sorry for the mistake in the numbering of the tables, we made the correction and inserted the correct numbering in the table.

17. Table 5 is not signed. Line 165: is it about Table 5?

Authors: Thanks for your careful review. Sorry for the mistake in the numbering of the tables, we made the correction and inserted the correct numbering in the table. Table 5 starts on pagne 16.

18. Table 6 is in a Supplementary Materials, line 271 – Table7. You need to correct the table number.

Authors: Thanks for your careful review. We performed the correction.

19. Line 214: (87.1 million). at 167.5) (59) – You should correct it.

Authors: Thanks for your careful review. We correct the information. Line: 354.

20. Line 238: Nam et al. (2020) – You should correct it: Nam et al. (68)

Authors: Thanks for your careful review. We correct the information. Line 378.

21. Line 240: Kaplan (2018) – Kaplan (69)

Authors: Thanks for your careful review. We correct the information. Line 380

22. The first limitation of this study is the fact that different tools were used in the assessment of back pain”. The limitation may be different periods of back pain and the lack of definition of back pain in many publications (see Table 2).

Authors: Thanks for your observation. Suggested limitations were inserted into the text, as follow:

Lines: 384-385 “…limitation may be different periods of back pain, lack of definition of back pain in many studies…”

Thank you very much. I congratulate the authors for their work. I think the article should be corrected and supplemented with missing information.

Thank you very much for all the necessary observations and for the opportunity to respond to each one of them.

Reviewer 2 Report

The authors conducted a systematic review and meta-analysis of studies investigating the prevalence and risk factors of musculoskeletal disorders in basketball players. While the topic is relevant and important, I am recommending that this paper be rejected due to several major issues with the methodology.

Firstly, the authors have not clearly described the inclusion and exclusion criteria for the studies included in the systematic review. This is a critical component of any systematic review and meta-analysis and without this information, it is difficult to assess the validity and appropriateness of the studies included in the analysis.

Secondly, there is no description of the data extraction process, which is essential to ensure the accuracy of the analysis. The authors should provide detailed information on how the data were extracted from the included studies, including the number of reviewers involved, the data extraction form used, and any discrepancies resolved.

Thirdly, there is no mention of the quality assessment of the included studies in the method section. The authors should describe the methods used to assess the quality of the studies, including any tools or criteria used.

Finally, the authors have not provided a clear description of the statistical analysis performed. It is essential to provide details on the statistical methods used, including the software and version, and any assumptions made.

In conclusion, the manuscript does not meet the basic standards for publication in a scientific journal. The methodology section lacks crucial information on the inclusion criteria, data extraction, quality assessment, and statistical analysis. 

Author Response

April 4, 2023.

Manuscript Number: Healthcare-2298321

Title: Prevalence and risk factors of musculoskeletal disorders in basketball players: systematic review and meta-analysis

Dear Editor,

We would like to thank you, the Editor, and the Reviewers for the thoughtful and in-depth comments in our manuscript. Your suggestions and remarks have helped us to reflect on our paper and improve it. We appreciate your commitment and effort. We have carefully considered every comment, promptly accepted all the suggestions, and made the alterations as recommended using the red color in the manuscript.

Please find below a point-by-point response to the Editors’ and Reviewers’ comments with answers in red font.

REVIEWER 2

The authors conducted a systematic review and meta-analysis of studies investigating the prevalence and risk factors of musculoskeletal disorders in basketball players. While the topic is relevant and important, I am recommending that this paper be rejected due to several major issues with the methodology.

Firstly, the authors have not clearly described the inclusion and exclusion criteria for the studies included in the systematic review. This is a critical component of any systematic review and meta-analysis and without this information, it is difficult to assess the validity and appropriateness of the studies included in the analysis.

Authors: Thanks for your observation. We are very sorry for the mistake, we believe that when transferring the text to the template (three pages of the original text were lost) we only identified the lack of these three pages through your observation. The text that did not appear for your review starts right after Table 1 and ends before the results section. Especially in item 2.2 Identification of relevant studies are the inclusion and exclusion criteria, according to the paragraphs below:

Lines 101 – 113: “Inclusion criteria were: a) basketball players of both sexes; b) age up to 50 years (articles with other age groups were included, provided the data were presented separately); c) observational studies (longitudinal, cross-sectional, cohort, and case-control); d) assessment of musculoskeletal disorders and back pain in basketball players (articles with other injuries were included, provided the data are presented separately); e) publications in English; and f) studies with players from communities of different nationalities.

Articles based on the following criteria were excluded: a) studies with basketball players that pertained to injuries in other body regions (knee, shoulder, hip, ankle) unless back and spine data were presented separately or could be calculated; b) paralympic athletes and/or players with physical or mental disabilities; c) mixed sports samples,, unless basketball player data were presented separately or could be calculated; d) experimental studies; and e) studies with incomplete data.”

Secondly, there is no description of the data extraction process, which is essential to ensure the accuracy of the analysis. The authors should provide detailed information on how the data were extracted from the included studies, including the number of reviewers involved, the data extraction form used, and any discrepancies resolved.

Authors: Thanks for your observation. Again, sorry for the same mistake Regarding item 2.4 Data Extraction, we added the paragraphs below:

Lines 151-157: “The following data were extracted from the selected studies: author and year of publication, type of study, country of origin, population, sex, age group, type of change, tool and prevalence of postural change and injuries, location and prevalence of back pain. The full description of the extracted data is included in Tables 2,3, 4 and 5.

We performed a meta-analysis with the data that were extracted to ascertain the prevalence of back pain, postural changes, and spinal injuries. However, these data were insufficient to perform a meta-analysis of the associated factors.”

Thirdly, there is no mention of the quality assessment of the included studies in the method section. The authors should describe the methods used to assess the quality of the studies, including any tools or criteria used.

Authors: Thanks for your observation. Regarding item 2.6 Methodological quality and risk of bias, we added the paragraphs below:

Lines 165-192: “The included articles were assessed for methodological quality and risk of bias by using the Grading of Recommendations, Assessment, Development, and Evaluations - GRADE [28] or the Downs and Black checklist [29]. GRADE is a method that serves transparency and simplicity by enabling the classification of the quality of evidence into four categories: very low quality, low quality, moderate quality, and high quality. GRADE has been adopted worldwide because of its rigorous methodological classification and ease of use [28].

An adapted version of the Downs and Black checklist that was proposed by Noll was used [30], wherein each item indicates: A) clearly stated objective; B) clearly described main outcomes; C) clearly defined sample characteristics; D) clearly described distribution of main confounders; E) clearly defined main findings; F) random variability in estimates provided; G) loss to follow-up described; H) probability values provided; I) representative target sample of the population; J) representative sample recruitment of the population; K) analyses adjusted for different follow-up times; L) properly used statistical tests; M) valid/reliable primary outcomes; N) sample recruited from the same population; O) adequate adjustment for confounders; and P) sample loss to follow-up considered (corresponding to items 1–3, 5–7, 9–12, 17, 18, 20, 21, 25, and 26). Items G and P were applied only to longitudinal studies, whereas items K and N were applied only to case–control and longitudinal studies. Scores reach 100% at 12, 14, and 16 points for cross-sectional, case–control, and longitudinal studies, respectively. Scores above 70% were used to define a low risk of bias [29]. The Downs and Black checklist was identified as one of the two tools that were most frequently used in systematic reviews which were registered in PROSPERO from 2011 to 2018 [31].

The scores were used to determine the methodological quality of the studies while considering five aspects: Presentation, external validity, internal validity - bias, internal validity - confounders, and statistical power for inferences [29]. Risk of bias was assessed independently by two examiners. The prevalence of data identified in the studies was used for the data synthesis strategy.”

Finally, the authors have not provided a clear description of the statistical analysis performed. It is essential to provide details on the statistical methods used, including the software and version, and any assumptions made.

Authors: Thanks for your observation. Here occurred the same mistake Regarding item 2.7 Statistical analysis, we added paragraphs below:

Lines 194-200: “Meta-analyses were performed to estimate the prevalence of back pain, spinal injuries, and postural changes. Data were presented graphically in Forest plots to estimate the prevalence rates with 95% confidence intervals (CI). Statistical I² values were calculated to quantify the degree of heterogeneity between studies, with values of 25–50% representing moderate heterogeneity and values >50% representing large heterogeneity among the studies [32]. Publication bias was assessed using Egger's test. All analyses were performed using STATA (version 16.0; StataCorp, College Station, TX, USA).”

In conclusion, the manuscript does not meet the basic standards for publication in a scientific journal. The methodology section lacks crucial information on the inclusion criteria, data extraction, quality assessment, and statistical analysis.

Authors: Sorry for the mistake, I believe that when transferring the text to the template (three pages of the original text were lost) we only identified the lack of these three pages through your observation. These three pages contained exactly the items pointed out by you (inclusion and exclusion criteria, data extraction, methodological quality and statistical analysis. We appreciate the opportunity to send the part of the text that was lost in the submission process.

Reviewer 3 Report

The article is very well written, congratulations to the authors.

I would like to ask the authors for an image of the evolution of the research on the articles, demonstrating how many were excluded and included at the end of the research.

The inclusion of this table could replace table 1

Author Response

April, 2023.

Manuscript Number: Healthcare-2298321

Title: Prevalence and risk factors of musculoskeletal disorders in basketball players: systematic review and meta-analysis

Dear Editor,

We would like to thank you, the Editor, and the Reviewers for the thoughtful and in-depth comments in our manuscript. Your suggestions and remarks have helped us to reflect on our paper and improve it. We appreciate your commitment and effort. We have carefully considered every comment, promptly accepted all the suggestions, and made the alterations as recommended using the red color in the manuscript.

Please find below a point-by-point response to the Editors’ and Reviewers’ comments with answers in red font.

REVIEWER 3

The article is very well written, congratulations to the authors.

Authors: Thanks.

I would like to ask the authors for an image of the evolution of the research on the articles, demonstrating how many were excluded and included at the end of the research.

The inclusion of this table could replace table 1.

Authors: Thanks for your careful review. I believe that there was an error in the processing of our file, because in our original article there is figure 1 which is the Preferred Reporting Items for Systematic Reviews and Meta-Analyses (PRISMA) flow diagram of articles included in this review, but only after your comment I noticed that figure 1 did not appear in the pdf made available to you. Therefore, we will provide for its inclusion again right after the second paragraph of the results section on Page 4.

Round 2

Reviewer 1 Report

Review Manuscript healthcare-2298321, entitled " Prevalence and risk factors of musculoskeletal disorders in basketball players: systematic review and meta-analysis."

Type of manuscript: Systematic Review

Journal: Healthcare

Dear Authors.

Suggestions have been included in the article. However, I have a couple of minor requests to be revised as stated below.

Comments:

1.     Lines 210-212: You should add references – “of which 14 investigated back pain […], 3 investigated postural changes […], 10 investigated spinal injures […], and 6 investigated back pain and spinal injuries […] (Table 2).

2.     Lines 64 - 66: “…aimed to evaluate the prevalence of non-specific back pain whose causes remain un-certain and probably multifactorial and musculoskeletal disorders in basketball players.” In a situation where the article evaluates back pain and injuries (such as spinal cord, nerve roots, bone structure, and disc ligaments of the spine – lines 96-98), I am not sure that the pain is non-specific. Studies should be excluded if they focused on low back pain associated with specific causes (e.g. back pain potentially associated with neurological compromise, suggested by the presence of nerve root compromise, sciatica, and back pain potentially associated with another specific spinal cause). Or in the methodological section (inclusion or exclusion criteria), explain what type of pain is included in this review.

3.     Line 87: PICO not PECO

Line 89: “C represents spine regions…” You should correct this sentence. Comparison: what is the alternative to the intervention – e.g. a different type of sports or a healthy population?

4.     Line 189: presentation not Presentation

5.     Lines 210-212 contain similar information as lines 226-228

6.     Line 226: [7,9,10,16,33,34,37,39,42,46–50,57],49–53,55) – You should correct it.

7.     Lines 231-231: 69.7% or n=23

Lines 233-234: 63.6% or n=21

There is no need to repeat the data. Check the whole text.

8.     Lines 218, 221, 234: number 59 and 60. Line 210: [6,7,9–11,16,33–58] – it is contradictory.

Line 249: Whether reference number 59 is included in this review? You should check it carefully. Check the whole text.

9.     Line 225: unnecessary space

10.  Lines 256-257: lesion?

11.  Line 250: (n=3; Table 3) – it should be corrected

12.  Lines 269-272: 5 articles reported back pain without indicating the specific location. Ok. That is clear, but the sentence needs to be rephrased.

13.  I repeat the question: For how long did the subjects experience back pain? The studies asked about the occurrence of LBP in the period of the last seven days, in the last month, three months, the last six months, the last year or life-time prevalence. Or studies did not include information in which period LBP occurrence was analysed. We know little about it. This issue could be developed in the Discussion section, not only in the limitations.

14.  The citation to Table 4 is missing. Maybe Table 4 is not needed. Certainly duplicates the information in Table 2.

 Thank you very much. I think the article should be corrected and supplemented with missing information.

Author Response

April 11, 2023.

Manuscript Number: Healthcare-2298321

Title: Prevalence and risk factors of musculoskeletal disorders in basketball players: systematic review and meta-analysis

Dear Editor,

We would like to thank you, the Editor, and the Reviewers for the thoughtful and in-depth comments in our manuscript. Your suggestions and remarks have helped us to reflect on our paper and improve it. We appreciate your commitment and effort. We have carefully considered every comment, promptly accepted all the suggestions, and made the alterations as recommended using the red color in the manuscript.

Please find below a point-by-point response to the Editors’ and Reviewers’ comments with answers in red font.

REVIEWER 1

Dear Authors.

Suggestions have been included in the article. However, I have a couple of minor requests to be revised as stated below.

Comments:

Comments:

1. Lines 210-212: You should add references – “of which 14 investigated back pain […], 3 investigated postural changes […], 10 investigated spinal injures […], and 6 investigated back pain and spinal injuries […] (Table 2).”

Authors: Thanks for your careful review. As per comment 5 that lines 210-212 and lines 226-228 had similar content. We reformulate it as follows, from line 210 we leave the more general information and remove the specific ones that are already well described from line 226 onwards. Page 5, Lines 208-209.

2. Lines 64 - 66: “…aimed to evaluate the prevalence of nonspecific back pain whose causes remain uncertain and probably multifactorial and musculoskeletal disorders in basketball players.” In a situation where the article evaluates back pain and injuries (such as spinal cord, nerve roots, bone structure, and disc ligaments of the spine – lines 96-98), I am not sure that the pain is non-specific. Studies should be excluded if they focused on low back pain associated with specific causes (e.g. back pain potentially associated with neurological compromise, suggested by the presence of nerve root compromise, sciatica, and back pain potentially associated with another specific spinal cause). Or in the methodological section (inclusion or exclusion criteria), explain what type of pain is included in this review.

Authors: The reviewer is absolutely right. Sorry for our mistake. We removed in the aim and also in the discussion section the mention on “non-specific” pain. Actually, we evaluated in a wide way, both specify or non-specific pain were included. To make it even clearer we also include the following snippet:

Both specify or non-specific back pain were included.” Page 3, Lines 105-106.

3. Line 87: PICO not PECO

Line 89: “C represents spine regions…” You should correct this sentence. Comparison: what is the alternative to the intervention – e.g. a different type of sports or a healthy population?

Authors: Thanks for your observation. Our intention really was to use PECO which is a variation of PICO. Because in our study we are not evaluating intervention, but exposure. For this reason, we state that according to the reference literature [19] the correct term to be used in this situation is the term PECO, as follows:

"P" represents players

"E" represents basketball sport

"C" represents spine regions (lumbar spine, cervical spine, thoracic spine)

"O" represents the prevalence of back pain and musculoskeletal disorders (postural changes and spinal injuries) in basketball players and associated factors

4. Line 189: presentation not Presentation

Authors: Thanks for your careful review. we performed the correction. Page 05, Line 188.

5. Lines 210-212 contain similar information as lines 226-228

Authors: Thanks for your observation. From line 210 we leave the more general information and remove the specific ones that are already well described from line 226 onwards. Page 5, Lines 208-209.

6. Line 226: [7,9,10,16,33,34,37,39,42,46–50,57],49–53,55) – You should correct it.

Authors: Thanks for your careful review. we performed the correction. Page 12, Line 224.

7. Lines 231-231: 69.7% or n=23 Lines 233-234: 63.6% or n=21 There is no need to repeat the data. Check the whole text.

Authors: Thanks for your observation. We standardize the information according to the rest of the text. Page 12, Lines 228 and 231.

8. Lines 218, 221, 234: number 59 and 60. Line 210: [6,7,9–11,16,33–58] – it is contradictory.

Authors: Thanks for your careful review. we performed the correction. Page 12, Line 209.

Line 249: Whether reference number 59 is included in this review? You should check it carefully. Check the whole text.

Authors: Thanks for your observation. Yes, reference 59 is included and we check the entire text.

9. Line 225: unnecessary space

Authors: Thanks for your careful review. Space has been removed.

10. Lines 256-257: lesion?

Authors: Thanks for your observation. We substituted the term "lesion" for "postural changes", which really is the most appropriate term. Page 15, Line 261.

11. Line 250: (n=3; Table 3) – it should be corrected

Authors: Thanks for your careful review. we performed the correction. Page 15, Line 255.

12. Lines 269-272: 5 articles reported back pain without indicating the specific location. Ok. That is clear, but the sentence needs to be rephrased.

Authors: Thanks for your observation. Based on your suggestion, we rewrote the sentence as follows:

Page 16, Lines 273-276: “In 80% (n=16) of the studies low back pain was identified [6,7,50,54-56,59,60,8,10,37,38,43,46,47,49]; in 25% (n=5) of the studies was reported back pain [8,34,42,49,58]. In 25% (n=5) of the studies more than one outcome was reported for the location of back pain [7,8,37,42,49].”

13. I repeat the question: For how long did the subjects experience back pain? The studies asked about the occurrence of LBP in the period of the last seven days, in the last month, three months, the last six months, the last year or life-time prevalence. Or studies did not include information in which period LBP occurrence was analysed. We know little about it. This issue could be developed in the Discussion section, not only in the limitations.

Authors: Thanks for your careful review. The following excerpt was included in the results and discussion sections:

Page 12, Lines 234-239: “The vast majority of studies, a total of 14, did not present the period in which low back pain was assessed [6,38-41, 43-46,48,52,55,57]. A total of 8 studies assessed low back pain in the last 3 to 6 months [33,35-37,42,51,56,59], 5 studies assessed it over a period of one year [10,16,34,47,50], 4 studies assessed it over a lifetime [7,9,54,60], and finally 2 studies assessed it within the last 3 to 5 five years [49,53].”

Page 22, Lines 390-393: “The first limitation of this study is the limitation may be different periods of back pain, for example, we have studies that did not mention the period in which the prevalence was verified [6,38-41, 43-46,48,52,55,57], and we also have studies that verified the prevalence in the last 3 months [33,42] up to the last five years [53]”.

14. The citation to Table 4 is missing. Maybe Table 4 is not needed. Certainly duplicates the information in Table 2. Thank you very much. I think the article should be corrected and supplemented with missing information.

Authors: Thanks for your observation. we included the citation to table 4 at the end of section 3.2 Main characteristics of the studies. Page 12, Lines 238-239.

Reviewer 2 Report

Based on my review of the revised version, I can confirm that the modifications you made have been executed correctly and have significantly enhanced the manuscript.

Author Response

April 11, 2023.

Manuscript Number: Healthcare-2298321

Title: Prevalence and risk factors of musculoskeletal disorders in basketball players: systematic review and meta-analysis

Dear Editor,

We would like to thank you, the Editor, and the Reviewers for the thoughtful and in-depth comments in our manuscript. Your suggestions and remarks have helped us to reflect on our paper and improve it. We appreciate your commitment and effort. We have carefully considered every comment, promptly accepted all the suggestions, and made the alterations as recommended using the red color in the manuscript.

Please find below a point-by-point response to the Editors’ and Reviewers’ comments with answers in red font.

REVIEWER 2

Based on my review of the revised version, I can confirm that the modifications you made have been executed correctly and have significantly enhanced the manuscript.

Authors: Thanks for your observation. We greatly appreciate the opportunity to make the corrections and also for your willingness to review them again.